# Cancer Therapy Targeting CD47/SIRPα

**DOI:** 10.3390/cancers13246229

**Published:** 2021-12-11

**Authors:** Nazli Dizman, Elizabeth I. Buchbinder

**Affiliations:** 1Department of Internal Medicine, Yale School of Medicine, New Haven, CT 06510, USA; nazli.dizman@yale.edu; 2Department of Medical Oncology, Dana-Farber Cancer Institute, 450 Brookline Ave, Boston, MA 02215, USA

**Keywords:** CD47, SIRPα, innate immunity, macrophage, immunotherapy

## Abstract

**Simple Summary:**

The interaction between cluster of differentiation 47 (CD47) on cancer cells and signal regulatory protein alpha (SIRPα) on immune cells, such as macrophages and dendritic cells, generates a “don’t eat me” signal. This is a common mechanism that provides cancer cells an escape from the innate immune system. Several therapeutics directed to CD47 or SIRPα have entered early clinical trials in recent years. In this article, we review the role of CD47/SIRPα axis in cancer, and summarize the literature on the efficacy and safety of therapeutics targeting CD47 or SIRPα. We also discuss the future implementation of these therapeutics in the treatments of various cancer types.

**Abstract:**

In the past decade, the field of cancer immunotherapy has rapidly advanced, establishing a crucial role for immune checkpoint blockers in the treatment of a variety of cancer types. In parallel with these remarkable clinical developments, further efforts have focused on ways of unleashing adaptive immune responses against cancer. CD47, a cell surface molecule overexpressed by several cancer types that facilitates immune escape from macrophages, dendritic cells and natural killer cells, and its ligand SIRPα, have emerged as potential therapeutic targets. A number of agents directed to CD47/SIRPα have been developed and demonstrated preclinical activity. Early phase clinical trials are investigating CD47/SIRPα directed agents with available data, suggesting safety and preliminary activity. Herein, we provide an overview of the mechanistic rationale of targeting CD47/SIRPα axis and associated clinical evidence.

## 1. Introduction

Immunotherapy with immune checkpoint blockade, cellular therapy and an emerging range of novel approaches to target the immune system are rapidly changing cancer care. The earliest immunotherapies utilized cytokines such as interleukin-2 and interferon, paving the road for subsequent discovery and advancement [1,2]. The revelation that immune checkpoint inhibition targeting cytotoxic T-lymphocyte associated protein-4 (CTLA-4) or programmed cell death protein (PD-1) could prolong survival in a range of malignancies launched a revolution in the immunotherapy space [3,4]. Numerous therapies have emerged and currently are in testing. These include small molecules, antibodies, modified viruses, and cellular therapies including tumor-infiltrating lymphocyte (TIL) therapy and chimeric antigen receptor (CAR) T cell therapies that have led to a profound advancement with substantial efficacy in hematologic malignancies and potential for activity in solid tumors [5,6,7]. Nevertheless, despite these remarkable advancements designed to expand the T cell-mediated immunity, the majority of cancer patients do not respond to or develop resistance to immunotherapy, highlighting the need for additional approaches to expand cancer immunotherapy [8].

Investigations have focused on identifying the processes by which evolving tumor cells overcome innate immunity. Among various pathophysiologic mechanisms fostering immune evasive tumor behavior, cluster of differentiation 47 (CD47), a transmembrane molecule commonly present on non-malignant hematologic cells, including red blood cells and thrombocytes, has emerged as a promising target. Several solid tumor types and hematological malignancies overexpress CD47, enabling immune escape from components of the innate immune system such as macrophages, dendritic cells, and natural killer (NK) cells via binding signal regulatory protein alpha (SIRPα); these effects lead to the disruption of direct tumor killing, and the resultant impairment in antigen presentation and T cell infiltration [9]. Studies of several cancer types have suggested prognostic properties of CD47 overexpression. A number of mechanistically different approaches targeting the CD47/SIRPα axis in order to potentiate the innate immune system have been developed and advanced from preclinical studies to early clinical trials. These include monoclonal antibodies, fusion proteins and bispecific antibodies, along with combination strategies, with most demonstrating encouraging safety and preliminary clinical activity in both hematologic malignancies and solid tumors. Herein, we present a comprehensive review of the rationale behind targeting the CD47/SIRPα axis and CD47/SIRPα-directed therapeutics in clinical development. 

## 2. Innate and Adaptive Immune Systems and Cancer

The majority of immunotherapies in use at this time target adaptive immunity by stimulating and activating T cells to recognize cancer cells [10]. T cell activation begins early in the lymph nodes, when antigen presenting cells (i.e., macrophages and dendritic cells) display tumor antigens in the major histocompatibility complex, which binds the T cell receptor. The activation of T cells via antigen presentation is a highly regulated process, controlled by numerous immune checkpoints requiring the involvement of various co-stimulatory and co-inhibitory molecular interactions. These positive and negative immune checkpoints determine whether the activation will occur [11,12,13]. By its nature, immune checkpoint inhibition works to overcome negative signals designed to protect “self” cells from immune attack. In addition to the currently approved molecular inhibitors of immune checkpoints, several other novel immunotherapy approaches are being investigated in preclinical and clinical settings, including CAR-T cells and TIL therapy, both of which are T cell products targeting specific proteins on the surface of malignant cells, aiming to augment adaptive immunity [7,14].

Innate immunity is the first line of defense against pathogens and other threats [10]. The innate immune system includes macrophages that phagocytose invaders, and are involved in antigen presentation. In cancer, increased infiltration of macrophages within and surrounding the tumor mass correlates with increased tumor invasiveness, growth, and poor prognosis. There is a correlation between tumor-associated macrophages (TAMs) and poor prognosis in breast, prostate, ovarian, and cervical cancers [15,16,17]. Clinical evidence of TAM participation in tumor growth is corroborated in animal models where macrophage signaling is inhibited or knocked out, suggesting that improving the response of the innate immune system to malignancy has potential to build on therapy options for cancer patients [18].

## 3. CD47/SIRPα Axis 

CD47, also known as integrin associated protein, is a transmembrane protein, belonging to the immunoglobulin superfamily broadly expressed on a variety of cell types [19]. CD47 has been identified as a marker on red blood cells where it serves as a “don’t eat me” signal to inhibit red blood cell phagocytosis [20]. As red blood cells age, they lose CD47 signaling, leading to the removal of older cells from circulation by macrophages in the spleen [20,21]. Notably, this overexpression is also seen on several other normal cells, including circulating hematopoietic stem cells and progenitor cells [9,19,20]. The ligands of CD47 include integrins, thrombospondin-1 (TSP-1), and SIRPα, a regulatory membrane glycoprotein expressed on various innate immune system cells, including macrophages and dendritic cells, along with granulocytes, monocytes and neurons [19,22]. SIRPα consists of an extracellular region with three Ig-like domains, and an intracellular region with two immunoreceptor tyrosine-based inhibitory motifs [23]. Upon activation via binding to CD47, the immunoreceptor tyrosine-based inhibitory motif regions become phosphorylated and induce the recruitment and activation of protein tyrosine phosphatases SHP-1 and SHP-2, leading to dephosphorylation of downstream molecules such as myosin IIA and repression of phagocytosis [24]. Overall, under physiologic conditions, CD47 and SIRPα interaction is one of the mechanisms protecting normal cells from macrophage-mediated phagocytosis [22].

## 4. Role of CD47/SIRPα in Cancer

In the early 1990s, the first oncological studies of CD47 identified it as a potential tumor marker for ovarian cancer [25]. This was followed by investigations on a wide variety of solid and hematological cancer types, including head and neck small-cell carcinoma (HNSCC), breast cancer, acute myeloid leukemia (AML), non-Hodgkin’s lymphoma (NHL), myeloma demonstrating differential overexpression of CD47 between cancer cells and matched normal cells [9,20,21,24,26,27,28,29].

The role of the CD47/SIRPα interaction in providing an escape mechanism for cancer cells from macrophage targeting has been well described. Human-derived xenograft models for several types of malignancies demonstrated sensitivity to CD47-blocking antibodies. In culture, these antibodies induced the macrophage-mediated phagocytosis of tumor cells [30,31,32,33,34,35,36,37]. The impact of the CD47 blockade on macrophage populations within the tumor microenvironment was also studied. In brief, TAMs display different polarization states between M1 macrophages with anti-tumor phenotypes and M2 macrophages with pro-tumor and immunosuppressive phenotypes [38,39,40]. In a human glioblastoma model, anti-CD47 therapy increased M1 macrophages within the tumor. This finding suggests that anti-CD47 therapy may play a role in shifting the phenotype of macrophages toward the anti-tumorigenic M1 subtype [41]. CD47 signaling also participates in macrophage recruitment into tumors. Weiskopf and colleagues showed that phagocytosis, following anti-CD47 treatment, causes systemic and local secretion of chemokines and cytokines that recruit macrophages into tumors in mice engrafted with small-cell lung cancer (SCLC) cell lines [30].

Beyond the activation of macrophage-mediated tumor killing, CD47-SIRPα interruption exerts other multidimensional positive effects on the immune response against cancer cells. For example, CD47-SIRPα blockade augments antibody dependent cellular cytotoxicity (ADCC) via the inhibition of SIRPα, expressed on the surface of NK cells [42,43]. Kim and colleagues demonstrated that impaired NK cell activity present in HNSCC cell lines overexpressing CD47 could be reversed with anti-CD47 antibodies [44]. CD47-SIRPα antagonist agents with an intact or even partially inactive Fc portion embedded in their structure may foster anti-tumor activity via antibody opsonization and destruction of target cells through ADCC or antibody-dependent cellular phagocytosis (ADCP) [45]. In addition, CD47/SIRPα interaction also has roles in tumor cell apoptosis, proliferation and migration [46,47,48]. CD47 inhibition can also negatively impact the function of other CD47 ligands, such as TSP-1 and integrins. These indirect effects may contribute to the anti-tumor and pro-inflammatory activity of CD47 inhibition. Despite contrasting evidence, a growing body of research highlights the role of TSP-1 in cell proliferation, invasion, metastatic potential, and worse survival rates, either through its interaction with CD47 or independently [49,50,51]. Notably, Kamijo and colleagues reported an association between high TSP-1 expression and worse disease-free survival in cutaneous T cell lymphoma patients. TSP-1 was found to be overexpressed in cutaneous T cell lymphoma, and anti-CD47 antibodies led to the inhibition of TSP-1-mediated cell proliferation in vivo [52]. 

Preclinical work has suggested a synergy between the cytotoxic agents and the CD47 inhibitors, especially when cytotoxic therapies were introduced prior to CD47-directed therapies. Neoantigens and nucleic acid remnants, produced from dying cancer cells and released into the tumor microenvironment after chemotherapy, may potentiate anti-CD47 activity [53]. In the context of hematologic malignancies, in vitro studies showed that azacytidine (a standard of care DNA hypomethylating agent used in the treatment of AML) and myelodysplastic syndrome and venetoclax (a B-cell lymphoma-2 inhibitor used in AML), induces the expression of other pro-phagocytic pathway components such as calreticulin and CD47 [54].

Perhaps more intriguingly, the macrophages involved in phagocytosis function as antigen-presenting cells, linking innate and adaptive immunity [29,53,55]. Thus, targeting the CD47-SIRPα axis, either through the CD47 or SIRPα blockade, may also promote antigen-presenting cell function, and stimulate T cell-mediated anti-cancer immunity (Figure 1) [56,57]. Studies in preclinical models with cancer types including chronic lymphocytic leukemia, colon cancer, melanoma, HNSCC, and glioblastoma, showed the induction of antitumor cytotoxic T cell populations, and reduced regulatory T cell populations in response to anti-CD47 treatment [26,55,58,59,60]. These observations were replicated in ex vivo studies. For example, Tao and colleagues assessed tumor samples from esophageal squamous cell cancer patients, showing an inverse relationship between CD8 T cell density and CD47 expression. In mice models with esophageal squamous cell cancer, treatment with anti-CD47 antibodies led to an increase in PD-1 and CTLA-4 expression. Treatment with the combination of CD47, PD-1 and CTLA-4 inhibitors yielded significantly improved survival in mice, compared with anti-CD47 monotherapy or PD-1 and CTLA-4 inhibitor combination, suggesting a rationale for combinatory therapeutic approaches to obtain synergistic effects [61].

Clinical implications of CD47 overexpression were also studied in various cancer types with the majority showing an inverse relationship between CD47 overexpression and clinical outcomes [62]. Chao et al. used flow cytometry and found that NHL cells had two-fold greater CD47 expression than normal germinal center and peripheral blood B cells. Grouping patient samples based on *CD47* mRNA expression levels, investigators showed improved overall survival in patients with CD47 low tumors, especially diffuse large B cell lymphoma (DLBCL), B cell chronic lymphocytic leukemia, and mantle cell lymphoma subsets [63]. Majeti and colleagues, showed high CD47 expression by gene expression arrays and flow cytometry in leukemia stem cells, compared with normal counterparts in a group of 137 AML patients. Compared with those with low CD47 expression, patients with high CD47 expression had significantly worse overall survival rates (22.1 vs 9.1 months, hazard ratio (HR): 2.02) and event free survival (17.1 vs 6.8 months, HR 1.94) [32]. Analyzing immunohistochemistry staining of CD47 in bone marrow biopsy samples from 248 AML patients, Galli et al. detected high CD47 staining in one-fourth of the patient samples. Samples with high CD47 staining had higher median blast count, median bone marrow infiltration, and disease burden. Although there was a trend towards unfavorable progression free survival in patients with high CD47, no statistical difference was observed in median progression-free survival, or overall survival [64]. Melanoma patients with tumors bearing CD47 overexpression were found to have worse overall survival rates and higher rates of distant metastasis [65]. Similarly, head and neck cancer patients with tumors bearing robust CD47 immunohistochemistry staining had diminished overall survival, compared with those with low CD47 staining [26]. A study of ovarian cancer demonstrated that increased CD47 expression is associated with worse prognosis, increased migration and invasion, and the induction of epithelial-mesenchymal transition [66].

## 5. Therapies Targeting CD47/SIRPα in Cancer

As a result of the promising preclinical data regarding the anti-tumor activity of CD47/SIPRα blockade obtained from in vivo and in vitro studies, several molecules have been developed and are undergoing clinical testing. Functionally, therapeutics under investigation may be classified as (1) CD47 targeting agents, (2) SIRPα targeting agents and (3) bispecific targeting agents. Table 1 provides a comprehensive list of the ongoing clinical trials of the CD47/SIRPα targeting therapeutics at the time of this publication. Although most of those approaches are currently being tested in early-phase clinical trials to assess safety and tolerability, available data from a number of published studies has revealed promising activity and favorable tolerability. In addition to being tested on their own, trials of combinations with other anti-tumor agents are underway. Inspired by the fact that CD47/SIRPα signaling limits the efficacy of tumor-opsonizing antibodies, a number of clinical trials are evaluating agents targeting this axis in combination with agents such as rituximab, cetuximab and trastuzumab [30,63,67]. Histone deacetylase (HDAC) inhibitors have been shown to enhance checkpoint inhibitor therapy by decreasing immune suppressive cells and increasing tumor antigen presentation [68,69]. Given the possible enhancement of tumor immunity, combinations of HDAC inhibitors and CD47 targeting therapies are underway. Other strategies employ a combination of CD47 targeted therapies with immune checkpoint inhibitors and chemotherapies.

Major concerns regarding the use of CD47-targeted agents are driven by the ubiquitous expression of CD47, which leads to rapid drug elimination, “antigen sink” and hematologic toxicity, such as anemia and thrombocytopenia [70]. The impact on hematopoietic cells, particularly red blood cells, presents a substantial issue with CD47/SIRPα targeted drugs. Given that older red blood cells are more sensitive to phagocytosis, red blood cell destruction remains a limiting toxicity with these drugs, and may influence the age of patients that can be treated with these agents [71]. Notably, as opposed to red blood cells, other normal functioning cells are less vulnerable to macrophage-mediated immune destruction with anti-CD47 therapies, as further activation of prophagocytic signals and involvement of calreticulin are suggested to be necessary steps to generate immune-related adverse events [72,73]. Furthermore, the use of CD47 or SIPRα-directed antibodies structured with an intact Fc portion raises similar concerns, due to the widespread expression of Fc receptors on normal cells, and the risks for potentially causing immune destruction of “self” cells [45]. From the drug development standpoint, the efficacy of the CD47/SIRPα blockade can vary based upon drug delivery method and compartmentalization of the drug. In addition, tumor type and stage, tumor immune microenvironment and acquired drug resistance, constitute other potential determinants of the efficacy of CD47/SIRPα inhibitors [74].

## 6. Anti-CD47 Antibodies and CD47-Targeting Recombinant Proteins

CD47-directed monoclonal antibodies and fusion proteins with SIRPα immunoglobulin structure competitively bind CD47 and block the interaction between CD47 and SIRPα. This class of therapeutics constitute the majority of the available in-human data testing CD47/SIRPα inhibition in solid tumors and hematologic malignancies, although data remains limited.

Hu5F9-G4 (5F9, magrolimab) is a humanized antibody with an IgG4 Fc fragment [75]. In a preclinical setting, magrolimab demonstrated anti-tumor activity against AML in-vitro and in vivo. Furthermore, complete disease elimination was observed in human B lymphoblastoid cell-engrafted mice, after treatment with magrolimab in combination with rituximab [75]. Preclinical models testing magrolimab in solid tumors such as colon, liver, ovarian and breast cancers demonstrated promising anti-tumor activity [76]. Another study in which patient-derived NHL xenografted mice were treated with magrolimab/rituximab combination showed an 89% cure rate, defined as over 4 months of disease free survival following the discontinuation of therapy [63]. In-depth analyses suggested that rituximab plays a complementary role in further stimulation of innate immunity, via its active Fc effector function-inducing natural killer cell and macrophage-mediated cellular cytotoxicity. Accordingly, the data from a phase Ib study of 22 patients with relapsed or refractory NHL, 95% of whom were previously treated with rituximab, demonstrated encouraging outcomes with an objective response rate of 50%, and a complete response rate of 36%, with magrolimab and rituximab in combination [77]. Adverse events experienced by patients on trial included chills, anemia and headaches (41% each), all of which occurred only in the first weeks of the trial. There were no significant safety signals in the latter stages of the trial. A simultaneously conducted phase I study of single agent magrolimab in metastatic solid tumors demonstrated a similar safety profile, with transient treatment-related adverse events [78]. Of note, trends in anemia development and transfusion requirements with magrolimab were further examined using the data from the patient population in the phase I dose escalation part of these studies [79]. Patients on escalating doses of magrolimab experienced a median 1.0 g/dL decrease in hemoglobin levels, and subsequent doses were associated with a lesser degree of hemoglobin decline. Red blood cell transfusion yielded appropriate responses in hemoglobin concentration, supporting the evidence regarding the transient nature of anemia after magrolimab administration [79]. A number of clinical trials evaluating magrolimab, either as a single agent or in combination with cytotoxic therapies, targeted therapies or immune checkpoint inhibitors to treat hematologic neoplasms, are ongoing. 

Other CD-47-targeting monoclonal antibodies that have entered clinical development include IBI188 (letaplimab), AK117 and SRF231. A phase I study of letaplimab in patients with advanced solid tumors and lymphomas was recently completed. Letaplimab demonstrated a favorable toxicity profile, with no dose-limiting toxicities. The majority of the treatment-related adverse events were grade 1–2. The rate of anemia was 15%, and only one patient developed grade 3 anemia. Notably, infusion related reactions were seen in 65% of the patient population, but all were grade 1–2 and manageable with a standard infusion-related reaction treatment algorithm [80]. AK117 monotherapy in patients with metastatic solid tumors demonstrated safety with no dose-limiting toxicities, no infusion-related reactions, or grade ≥ 3 treatment-related adverse events observed with up to 20 mg/kg dosing. Further dose escalation is underway with 30 mg/kg dosing [81]. For SRF231, further exploration in clinical trials was held by the pharmaceutical company. 

SIRPα-Fc fusion proteins are produced via combining the CD47 binding domain of SIRPα and the Fc region of human IgG1 or IgG4. These comprise a mechanistically different group of novel therapeutics, that are targeting CD47. SIRPα-Fc fusion proteins undergoing phase I clinical trials were engineered using different Fc structures to balance benefit and toxicity. Therefore, the extent of the contribution from the Fc portion of the fusion proteins on Fc receptor engagement, ADCC, ADCP and overall anti-tumor activity, may vary across strategies [45]. A SIRPα-Fc fusion protein, TTI-621 which is engineered using the Fc region of IgG1, competes with endogenous SIRPα in immune cells [82]. This agent can trigger phagocytosis of tumor cells by subsets of isolated TAMs. In the phase I study of TTI-621, the maximum tolerated dose was safely determined in 18 patients, based on transient grade 4 thrombocytopenia; importantly, however, no clinically significant anemia events were observed [82]. TTI-621 was administered to 146 patients in the dose expansion phase as a single agent in patients with T cell non-Hodgkin lymphoma, and in combination with rituximab in relapsed or treatment refractory patients with B-cell NHL, or with nivolumab in patients with Hodgkin lymphoma [83]. Notable adverse events included infusion reactions and thrombocytopenia (in 20% of the population)—although none were dose-limiting. Encouraging objective response rates were observed with monotherapy in DLBCL (29%), T cell NHL (25%) and with TTI-621 and rituximab combination in DLBCL (21%). In patients with relapsed/refractory mycosis fungoides and Sézary Syndrome, the intratumoral injection of TTI-621 showed appreciable improvement in both injected and non-injected adjacent lesions in 89% of the patients, as well as a reduction in lesion severity in one of two patients on maintenance intralesional therapy, suggesting a potential for abscopal or systemic effect [84]. TTI-622 is a new SIRPα-Fc fusion protein variant structured using the Fc region of human IgG4. This structure was designed to augment prophagocytic function of CD47 inhibition, by enabling higher levels of medication due to lower Fc receptor interactions, and associated toxicities expected with IgG4 compared with IgG1. This agent is currently being tested in a phase I setting in NHL patients. Initial results from this study showed that TTI-622 was well tolerated with no dose-limiting toxicities and grade ≥3 thrombocytopenia or anemia [85]. Another engineered protein, CV1-hIgG4, demonstrated activity but resulted in notable toxicity in vivo [86]. 

ALX148 was constructed by combining the high affinity D1 domain of SIRPα with the inactive human IgG1 Fc domain to avoid toxicity [87,88]. Indeed, a phase I clinical trial of this agent as monotherapy confirmed its safety, with no dose-limiting hematologic or non-hematologic toxicities observed. Investigators proceeded with testing two 10 mg/kg and 15 mg/kg weekly doses of ALX148, in combination with rituximab in relapsed/recurrent NHL patients, yielding an objective response rate of 40% and 55%, respectively, and a favorable toxicity profile, with the only grade 3 toxicities being a decreased neutrophil count (6%) and anemia (3%) [89]. A more extensive investigation of this agent was recently completed in combination with trastuzumab, with or without chemotherapy in metastatic gastric/gastroesophageal junction cancers, and in combination with pembrolizumab, with or without chemotherapy in metastatic head and neck squamous cell carcinoma patients [90]. The safety profile was favorable, with low rates of cytopenias in all cohorts. Encouraging activity signals have emerged with ALX148, in combination with pembrolizumab in HNSCC patients, who were previously treated with ≥2 lines of therapy with an objective response rate of 40%, and the ALX148 and trastuzumab combination in gastric cancer patients who previously progressed on HER2-targeted therapies with an objective response rate of 21%. Accordingly, various combination strategies with ALX148 are being investigated in phase II settings, in patients with HNSCC and gastric/gastroesophageal junction cancer. This type of approach attempts to limit the toxicity on normal cells; however, the anti-tumor efficacy may be compromised by losing the ADCC and ADCP effect [88].

## 7. SIRPα Targeting Agents

Agents targeting SIRPα include anti-SIRPα antibodies and modified CD47 proteins targeting SIRPα. Given that SIRPα expression is limited to myeloid cells and neurons, this approach is attractive because it avoids broader binding and undesirable effects observed with agents targeting CD47 on normal cells. However, the use of SIRPα-directed therapies may be limited by binding to neurons expressing SIRPα [91,92]. In addition, there is concern about cross reactivity with other SIRP family members such as SIRPβ which is, as opposed to SIRPα, proinflammatory in nature and induces neutrophil migration and macrophage phagocytosis, and SIRPγ which has previously been shown to induce T cell migration [93,94,95,96].

Two anti-SIRPα monoclonal antibodies have recently entered early clinical trial testing: BI 765063 and GS-0189 (also known as FIS-189). Both agents were designed to bear high affinity to SIRPα without binding to SIRPγ [97,98,99]. The dose escalation part of the phase I study of BI 765063 monotherapy in metastatic tumors was recently published and showed an impressive safety profile with no dose-limiting toxicities. As expected, no anemia or thrombocytopenia were observed. The most common treatment-related adverse events were infusion-related reaction (46%) and fatigue (12%) [99]. The phase I study of BI 765063 is currently recruiting patients with advanced solid tumors for the dose expansion phase, where patients are randomized into either BI 765063 monotherapy or in combination with the PD-1 inhibitor. GS-0189 is being tested in the phase I setting as a single agent, or in combination with rituximab in relapsed or refractory NHL patients. 

Several other SIRPα-directed agents are currently in preclinical development [100,101,102,103]. For example, ADU-1805, an anti-SIRPα monoclonal antibody, generated to block all known SIRPα alleles to excel anti-tumor activity across SIRPα variants, demonstrated in vitro and in vivo activity [100]. KWAR23 is an anti-SIRPα antibody, which has been demonstrated to synergize with rituximab in mouse models of lymphoma [36]. While the effect was limited when KWAR23 was used on its own, the combination augmented myeloid cell-dependent killing. There has also been some synergy observed with KWAR23 and cetuximab or panitumumab in colorectal adenocarcinoma cell lines [36]. A novel high affinity CD47 variant, velcro-CD47, was developed with augmentation of the existing contact interference via N-terminal peptide extension leading to enhanced SIRPα binding. The velcro-CD47 demonstrated anti-tumor activity and synergy with tumor-specific monoclonal antibodies (trastuzumab or cetuximab) in tumor cell lines [101].

## 8. Bispecific Agents Targeting CD47 and Another Molecule

Bispecific agents comprise a novel group of recombinant antibodies designed to target two different cell surface molecules simultaneously. In addition to CD47/SIRPα blockade, bispecific agents currently under investigation were designed to target either another cancer specific cell surface molecule concurrently expressed on the target tumor cells (i.e., CD19, CD20, PD-L1) or a cell surface molecule expressed by T cells (i.e., PD-1). While the first strategy aims to improve specificity of tumor targeting and prevent off-target effects, the latter promises synergistic activity with stimulation of both innate and adaptive immune systems [104]. The primary nuances that require attention in developing the dual targeting CD47/SIRPα axis-directed bispecific agents include the degree of the CD47 binding capacity and the presence of the Fc portion. The intact Fc portion fosters immune effector cell functions, and provides a longer half-life to the engineered antibody model; however, this moiety may be associated with unwanted side effects, due to the expression of Fc receptors on normal cells resulting in a similar phenomenon, as observed with CD47 [105].

HX009, a PD-1 and CD47 dual blocker (via IgG4-Fc of anti-PD-1 antibody and extracellular domain of SIRPα) was assessed in a phase I setting, and demonstrated to be safe by achieving a maximum tolerated dose, without dose-limiting toxicities or hematologic adverse events in patients with metastatic solid tumors. Clinical benefit signals were observed in this heavily pretreated patient population, with a median of three previous treatment lines. Of 18 patients with follow-up imaging studies available, three patients achieved partial response, and seven patients achieved stable disease. A phase II study of HX009 is currently recruiting metastatic solid cancer patients [106].

IBI322, an anti-CD47/PD-L1 bispecific monoclonal antibody, was designed with a monovalent CD47-binding domain and bivalent PD-L1-binding domain, to enable selective tumor targeting and spare red blood cells [107,108]. As a result, IBI322 exerts 14-fold lower affinity to red blood cells, compared with anti-CD47 monoclonal antibodies, while enabling equivalent affinity to PD-L1 as anti-PD-L1 agents [109]. This promising agent is being investigated in large early phase clinical trials, in patients with metastatic solid tumors and hematologic malignancies separately.

SL-172154 is a bifunctional fusion protein, developed via a mechanistically novel technology; it consists of human SIRPα and CD40L parts, that are connected via a human Fc. Whereas SIRPα inhibits CD47, CD40L binds and activates CD40, a costimulatory molecule present on B cells, macrophages and dendritic cells [110]. CD40/CD40L interaction partakes in both innate and adaptive anti-tumor immunity via stimulating pro-inflammatory cytokine production, including IL12 and TNF- α, the induction of ADCC, and T cell-mediated immunity and antibody production. It is also currently an active area of investigation for drug development in immune-oncology [111,112,113]. In vivo studies of SL-172154 have demonstrated superior anti-tumor activity with the fusion protein over either CD47 antagonist, CD40 agonist antibody as monotherapy or the combination of the two, suggesting a synergistic role [110]. Moreover, experiments combining SL-172154 and CTLA-4 or PD-1 blockage yielded enhanced the survival benefit in mice. Intravenous and intratumoral formulations of SL-172154 are being studied in phase I clinical trials in patients with ovarian cancer and HNSCC, respectively. 

Several other dual blockers were generated with the goal of co-engaging two concurrently expressed tumor cell surface molecules. In light of the synergy observed with anti-CD47 and anti-CD20 combination in vivo, scientists generated bispecific antibodies (i.e., IMM0306 and CPO107) which target cells with CD47 and CD20 co-expression [63,105,114]. A similar strategy was employed for the development of TG-1801, but targeting CD19 instead of CD20. This design not only addresses the challenges raised by non-specific CD47 targeting and the antigen sink, observed with solely CD47 targeting agents, but also exploits the pro-phagocytic activity of engaged Fc receptors via CD19 and CD20 inhibition [105,114]. Promising response and survival outcomes were observed with IMM0306 in animal models with NHL. TG-1801 demonstrated pre-clinical efficacy in cell lines and mouse models of numerous B-cell malignancies. Co-administration with rituximab led to significant tumor-growth inhibition and regression [115].

## 9. Other Approaches Targeting CD47/SIRPα Axis

Several other mechanistically unique CD47/SIRPα-directed strategies are currently in the preclinical setting. A revolutionary advancement in immune-oncology research, CAR-T cell treatments were applied in the CD47 blockade arena, with studies showing tumor killing against lung, ovarian and pancreatic cell lines, as well as the blocking of pancreatic and ovarian cancer xenograft growth in vivo [116,117,118].

Novel drug delivery systems, such as nanomedicine and synthetic biology technologies, have entered preclinical development of anti-CD47 therapeutics as well. HuNb1-IgG4, a nanobody generated via fusion of heavy-chain IgG4 and anti-CD47 antibodies, offers promise with its small and stable molecular structure, high affinity to CD47 resulting in in vivo and in vitro activity against ovarian and lymphoma cancer, and lack of hematopoietic cytotoxicity [119]. Moreover, nanoparticles made up of multi-functionalized iron oxide nanoparticles that include the anti-CD47 antibody and gemcitabine have been developed for use in pancreatic cancer therapy. Chen et al. developed a bio-responsive fibrin gel solution, containing CD47-conjugated nanoparticles as a post-surgical tool to induce the local phagocytosis of cancer cells and modulate an innate and adaptive immune response [120]. Quorum-sensing bacteria that delivers single chain antibody fragments targeting CD47 has also been tested with increased anti-tumor immunity and reduced progression [121].

## 10. Conclusions

Immunotherapies to date have mainly targeted adaptive immunity. However, innate immunity plays an important role in controlling tumor development and growth. Macrophages have demonstrated a correlation between poor prognosis and the degree of anti-tumor immune activity. Modifying macrophages and other innate immune activity via the blockade of the CD47/SIRPα axis has emerged as an attractive approach in the field of hematology-oncology. Many mechanistically and functionally unique compounds have been developed and are currently undergoing clinical testing. Although only a small number of clinical trials have been published to date, available data suggests overall safety, along with considerable activity signals. Potential limitations of targeting the CD47/SIRPα axis includes the ability to target agents into the tumor and toxicity such as cytopenias, which may potentially be overcome by novel therapeutic strategies. Questions remain on the cancer-type specific efficacy of this approach, and the synergistic potential with chemotherapy, targeted therapy and immune checkpoint blockade. Promising early signals in combination with tumor-opsonizing antibodies suggest that this may be a way to expand the effectiveness of these drugs. In addition, there remains interest in the combined targeting of the innate and adaptive immune systems. Numerous ongoing clinical trials and further clinical investigations will guide the tumor type, optimal combination therapy and timing of the therapy targeting the CD47/SIRPα axis.

## Figures and Tables

**Figure 1 cancers-13-06229-f001:**
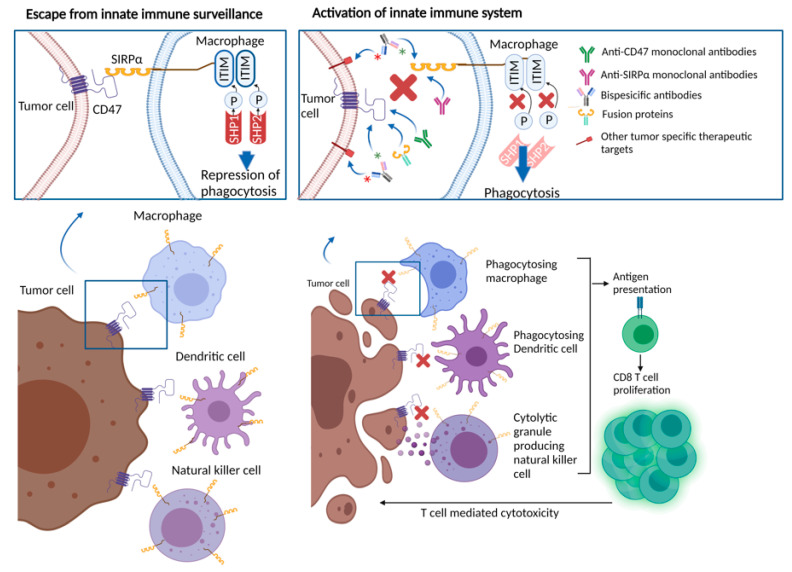
CD47/SIRPα interaction leading to repression of phagocytosis and therapeutic approaches blocking CD47/SIRPα axis.

**Table 1 cancers-13-06229-t001:** Clinical trials testing agents targeting CD47/SIRPα axis.

Agent	Therapeutic Target	Design	Phase	Disease Site	Accrual Goal	Identifier
**Monoclonal Antibodies**						
IBI188 (Letaplimab)	CD47	IBI188 +/− rituximab	I	Metastatic solid tumors or lymphoma	92	NCT03717103
		IBI188 +/− azacitidine	I	Myelodysplastic syndrome	12	NCT04485065
Hu5F9-G4 (Magrolimab)	CD47	Hu5F9-G4 (Magrolimab) + Pembrolizumab	II	Hodgkin’s lymphoma	24	NCT04788043
		Hu5F9-G4 (Magrolimab)	I	Hematologic malignancies	20	NCT02678338
		Hu5F9-G4 (Magrolimab) + acalabrutinib + rituximab or other combinations without Hu5F9-G4 (Magrolimab)	I	Non-Hodgkin’s Lymphoma	30	NCT03527147
		Hu5F9-G4 (Magrolimab) + Obinutuzumab + venetoclax	I	Non-Hodgkin’s Lymphoma	76	NCT04599634
ZL-1201	CD47	ZL-1201	I	Metastatic solid tumors or refractory lymphomas	66	NCT04257617
STI-6643	CD47	STI-6643	I	Metastatic solid tumors	24	NCT04900519
CC-9002	CD47	CC-90002 +/−rituximab		Part A: Metastatic solid tumors, multiple Myeloma or non-Hodgkin’s lymphoma Part B, relapsed and/or refractory CD20-positive NHL	60	NCT02367196
AK117	CD47	AK117	I	Metastatic solid tumors or lymphoma	162	NCT04728334
		AK117 + azacitidine	I/II	Myelodysplastic syndrome	190	NCT04900350
AO-176	CD47	AO-176 +/− paclitaxel	I/II	Metastatic solid tumors	132	NCT03834948
		AO-176 +/− dexamethasone or dexhamethasone + bortezomide	I	Multiple myeloma	102	NCT04445701
IMC-002	CD47	IMC-002	I	Metastatic solid tumors or lymphoma	24	NCT04306224
TQB2928	CD47	TQB2928	I	Metastatic solid tumors or hematologic malignancies	20	NCT04854681
FSI-189	SIRPα	FSI-189 +/− rituximab	I	Non-Hodgkin’s lymphoma (B-cell)	63	NCT04502706
BI 765063	SIRPα	BI 765063 +/− PD-1 inhibitor	I	Metastatic solid tumors with SIRPα polymorphism	116	NCT03990233
**Bispecific antibodies**						
HX009	CD47 and PD-1	HX009	II	Metastatic solid tumors	210	NCT04886271
PF-07257876	CD47 and PD-L1	PF-07257876	I	Non small-cell lung cancer, head and neck squamous cell carcinoma, ovarian cancer	90	NCT04881045
CPO107 (JMP601)	CD47 and CD20	CPO107 (JMP601)	I	Non-Hodgkin’s lymphoma (CD-20 positive)	75	NCT04853329
IBI322	CD47 and PD-L1	IBI322	I	Hematologic malignancies	182	NCT04795128
		IBI322	Ia	Metastatic solid tumors	45	NCT04338659
		IBI322	Ia/Ib	Metastatic solid tumors	218	NCT04328831
SL-172154	SIRPα and CD40L	SL-172154 (intravenous)	I	Ovarian cancer	40	NCT04406623
		SL-172154 (intratumoral)	I	Head and neck or cutaneous squamous cell carcinoma	18	NCT04502888
TG-1801	CD47 and CD19	TG-1801 +/− ubitixumab	Ib	Hematologic malignancies	60	NCT04806035
IMM0306	CD47 and CD20	IMM0306	I	Refractory or Relapsed CD20-positive B cell Non-Hodgkin’s Lymphoma	131	NCT04746131
**Fusion proteins**						
TTI-622	CD47 via SIRPαFc (IgG4) structure	TTI-622 + rituximab, PD-1 inhibitor, Proteasome inhibitor regimen or rituximab	Ia/Ib	Lymphoma or myeloma	156	NCT03530683
ALX148	CD47 via SIRPαFc (IgG1) structure	ALX148 + azacitidine	I/II	Myelodysplastic syndrome	173	NCT04417517
		ALX148 + venetoclax or azacitidine	I/II	Acute myleoid leukemia	97	NCT04755244
		ALX148	II	Head and neck squamous cell carcinoma	112	NCT04675333
		ALX148 + pembrolizumab	II	Head and neck squamous cell carcinoma	111	NCT04675294

## Data Availability

The data presented in this study are available on request from the corresponding author.

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
