# Peer review of "Cancer Therapy Targeting CD47/SIRPα"

_cancers, 2021, doi:10.3390/cancers13246229_

Round 1
Reviewer 1 Report
The authors significantly improved the manuscript with discussion about combinations and side-effects.
Minor points:
- The author field still has an extra "and" in Elizabeth Buchbinder's name.
- Page 6 line 269 "CD-47 targeting" should be "CD47-targeting"
Reviewer 2 Report
The authors added several sections in response to the critical comments of the reviewers making the manuscript greatly improved. No further comments are forthcoming.
This manuscript is a resubmission of an earlier submission. The following is a list of the peer review reports and author responses from that submission.
Round 1
Reviewer 1 Report
This is a well-written and timely review of a novel class of immunotherapeutic reagents targeting the CD47/SIRPa pathway which is responsible for cellular phagocytosis. Over-expression of CD47 can inhibit both innate and adaptive immune responses to tumor and as such, is a immune evasive mechanism to support tumor growth and spread. The review attempts to describe current efforts to inhibit this pathway in order to provide better immunotherapeutic responses in patients with different histological classifications. In addition, off-target binding and resulting toxicities are adequately discussed. This reviewer would like to point out a few issues that need to be addressed by the authors to clarify or correct the manuscript including the following:
- Although the role of SIRPa is well described, how do the other ligands of CD47 (ntegrins and thrombospondin-1) contribute to the off-target effects of ant-CD47 immunotherapy? Please elaborate.
- As stated by the authors, CD47 and SIRPa interactions protect normal cells from autoimmunity (Lines 95-6), so why do investigators only see relatively minor toxicities in patients undergoing this form of treatment?
- Lines 163-166 seem contradictory. Did the authors mean this or was this in error? If not in error, please try and explain.
- In the opinion of the authors, are the off-target effects of SIRPa binding agents to myeloid cells and neurons limiting the use of this approach and favoring therapies that target CD47?
- Line 323 and reference 92 describing velcro-CD47 should be more completely described so readers know how it is different than other CD47 reagents.
- Since tumors contain between 30-70% necrotic regions, does ant-CD47 affect binding of APCs to necrosis which could enhance immunological activation to tumor? Please comment on this.
- In Table 1, one type of combination therapy was not mentioned in the text. Specifically, the rational for combining HDAC inhibitors such as azacitidine or venetoclax and anti-CD47 immunotherapy needs clarification.
- Minor issues: line 32: exchange the word "targeting" with designed": Line 408: correct split infinitive "to effectively target"; Lines 399-400 reads as a run-on sentence. Please correct these.
Reviewer 2 Report
Major comments:
1) This is a well-written review article that summarizes history, expression, antibody/protein candidates, as well as concerns related to CD47. However, there are several papers published recently that has covered very similar materials (PMID: 34584838, 34609596, 34305918). In order for this review paper to provide novel insights to supplement previously available summaries to individual CD47 agents, the authors can consider expanding on the clinical implications, future combinations, and when to treat.
2) Please check author names (remove "and")
Minor comments:
Double check already defined abbreviations (example: TAMs were defined but full name appeared again later). There are double spaces on several occasions.
